# Gastric Cancer: An Up-to-Date Review with New Insights into Early-Onset Gastric Cancer

**DOI:** 10.3390/cancers16183163

**Published:** 2024-09-15

**Authors:** Marek Mazurek, Monika Szewc, Monika Z. Sitarz, Ewa Dudzińska, Robert Sitarz

**Affiliations:** 1Department of Surgical Oncology, Masovian Cancer Hospital, 05-135 Wieliszew, Poland; marekmazurek1@o2.pl; 2Department of Normal, Clinical and Imaging Anatomy, Medical University of Lublin, 20-950 Lublin, Poland; m.szewc456@gmail.com; 3Department of Conservative Dentistry with Endodontics, Medical University of Lublin, 20-950 Lublin, Poland; m.sitarz@umlub.pl; 4Department of Dietetics and Nutrition Education, Medical University of Lublin, 20-950 Lublin, Poland; ewa.dudzinska@umlub.pl; 5Department of Surgical Oncology, St. John’s Cancer Center, 20-090 Lublin, Poland

**Keywords:** early-onset gastric cancer, risk factors, therapy

## Abstract

**Simple Summary:**

Gastric cancer (GC) is ranked fifth among the most frequently diagnosed cancers and the fifth most common cause of cancer death in the world. We can classify cases of GC depending on the age at which it is diagnosed into early-onset GC (EOGC—up to the age of 45) and conventional GC (patients older than 45). Genetic factors are considered a likely cause of EOGC, as young patients are less exposed to environmental carcinogens. This comprehensive study presents all aspects: epidemiology, risk factors, new treatment strategies, and future directions.

**Abstract:**

Gastric cancer (GC) is the fifth most frequently diagnosed cancer and the fifth most common cause of cancer death in the world. Regarding the age at which the diagnosis was made, GC is divided into early-onset gastric cancer (EOGC—up to 45 years of age) and conventional GC (older than 45). EOGC constitutes approximately 10% of all GCs. Numerous reports indicate that EOGC is more aggressive than conventional GC and is often discovered at an advanced tumor stage, which has an impact on the five-year survival rate. The median survival rate for advanced-stage GC is very poor, amounting to less than 12 months. Risk factors for GC include family history, alcohol consumption, smoking, *Helicobacter pylori*, and Epstein–Barr virus infection. It has been shown that a proper diet and lifestyle can play a preventive role in GC. However, research indicates that risk factors for conventional GC are less correlated with EOGC. In addition, the unclear etiology of EOGC and the late diagnosis of this disease limit the possibilities of effective treatment. Genetic factors are considered a likely cause of EOGC, as young patients are less exposed to environmental carcinogens. Research characterizing GC in young patients is scarce. This comprehensive study presents all aspects: epidemiology, risk factors, new treatment strategies, and future directions.

## 1. Introduction

Gastric cancer (GC) ranks fifth in the world in terms of morbidity and mortality [1]. Regarding the age at which the diagnosis is made, GC is divided into early-onset GC (EOGC—up to the age of 45) and conventional GC (older than 45) [2]. EOGC constitutes approximately 10% of all GCs. There are no clear data on the etiology, histopathology, risk factors, and genome characteristics of EOGC. This makes EOGC a crucial yet challenging model to study. Presumably, genetic factors are the main reason for its formation because young patients are less exposed to environmental carcinogens [2,3]. Some authors report that EOGC is more aggressive than conventional GC, and the majority of young patients present at an advanced stage, which affects the five-year survival rate. The median survival rate in this case is very poor [4,5,6] and is less than 12 months [6,7]. The high aggressiveness and heterogeneous nature of EOGC still constitute a global health problem. Therefore, alternative prevention, early diagnosis, and proper follow-up treatments can lead to a reduction in recorded incidents [8].

## 2. Epidemiology

Several factors have a significant impact on the increased risk of developing GC, such as family history, diet (low in fruits and vegetables; rich in foods preserved via salting and processed and grilled meat), alcohol consumption, smoking, and *Helicobacter pylori* (*H. pylori*) and Epstein–Barr virus (EBV) infections [1,9], which are summarized in Figure 1. It has been postulated that the rising prevalence of autoimmune gastritis and dysbiosis of the gastric microbiome, probably related to the increased use of antibiotics and acid suppressants, may have contributed to the paradoxical increase in the incidence of GC among younger generations [10]. The influence of the use of proton pump inhibitors (PPIs), statins, nonsteroidal anti-inflammatory drugs (NSAIDs), or metformin on the development of EOGC hasalso been analyzed. However, no significant relationship between these factors and the disease was found. In contrast, the use of a cyclooxygenase-2 (COX-2) inhibitor for six months or longer before cancer diagnosis has been shown to significantly reduce the risk of EOGC [11].

Evidence suggests an etiological role for exposure to risk factors in early life and early adulthood. Since the mid-20th century, substantial multigenerational changes in the exposome have occurred (including changes in diet, lifestyle, obesity, environment, and the microbiome), which might interact with genomic and/or genetic susceptibilities [10]. As research indicates, genetic susceptibility factors may be associated with the development of GC [12].

The most pivotal indicator placing patients at a higher risk of developing GC is having a family history of this disease. Despite this, most GC cases are sporadic, with only around 10% showing a familial connection, and a hereditary cause is determined in 1–3% of cases [13,14,15]. No significant differences were observed in the occurrence of familial GC in patients with EOGC compared to late-onset GC patients [16]. Currently, the following hereditary syndromes are well characterized: hereditary diffuse gastric cancer (HDGC), familial intestinal gastric cancer (FIGC), gastric adenocarcinoma, and proximal polyposis of the stomach (GAPPS). The most noticeable familial GC is HDGC, a cancer induced by modifications in the gene coding E-cadherin (CDH1) [17].

GC may develop into other hereditary cancer syndromes, such as Li–Fraumeni syndrome (LFS), familial adenomatous polyposis (FAP), Peutz–Jeghers syndrome (PJS), Lynch syndrome (LS), hereditary breast/ovarian cancer syndrome (HBOCS), MUTYH-associated adenomatous polyposis (MAP), juvenile polyposis syndrome (JPS), and Cowden syndrome (CS). When comparing patients who had cases of gastric cancer among their close relatives with patients who did not, it was shown that a family history of GC increases the risk of developing this type of carcinoma by three times [17]. There is a stronger correlation between a family history of GC and the incidence of GC in Asia than in Europe or North America. However, even in Asia, there is no visible connection between a family history of HDGC and the occurrence of HDGC [18]. Thus, environmental factors play a more crucial role in the development of familial GC compared to genetic modifications.

Many studies have focused on the correlation between nutritional aspects and the risk of GC development. Potential carcinogens are activated by phase I enzymes [e.g., cytochrome P450 (CYP)] or detoxified by phase II enzymes [e.g., glutathione S-transferases superfamily (GST), N-acetyltransferase (NAT), and sulfotransferase (SULT)]. Genetic polymorphisms of these enzymes may alter their activity, and thus, affect the susceptibility to GC. The consumption of salt and salted food is known to elevate the risk of GC, either directly through damage to the gastric mucus or indirectly through correlation with *H. pylori* infection. In South Korea, a high consumption of kimchi and soybean pastes fermented with salt and other chemicals is a risk factor for GC among GSTM1-positive and GSTT1-positiveindividuals, as well as those carrying CYP1A1 Ile/Ile and CYP2E1 c1/c1, or with ALDH2 *1/*1 genotypes. Cooking meat at high temperatures for prolonged periods produces several potent carcinogens, including heterocyclic amines (HCAs) [19]. Boccia et al. reported that the positive association between the risk of GC and high consumption of grilled/barbecued meat is more pronounced among SULT1A1 His/His carriers than Arg/Arg carriers [19].

Mutagens found in ingested food may interact with gastric epithelial cells and induce alterations in genes and their expression. For instance, a high sodium chloride intake is said to be destructive to the gastric mucosa and amplifies cell apoptosis. The proposed mechanisms by which salt can cause GC are either direct damage to the gastric mucosa, leading to hyperplasia of the gastric pit epithelium, with increased potential for mutations, or an effect of interaction with *H. pylori*, as damage caused by salt may also increase *H. pylori* colonization of the stomach [20].

Both the dietary and endogenous roles of N-nitroso compounds significantly amplify the risk of gastrointestinal cancer, mostly among non-cardiac GCs. N-nitrosodimethylamine is classified as probably carcinogenic to humans (group 2A) by the International Agency for Research on Cancer. The major dietary sources of N-nitrosodimethylamine are cured meats, pickled fish and vegetables, and food products dried using direct-fire methods. Beer is also an important source of N-nitrosodimethylamine [21].

The habits of individuals that play a role in GC development have been examined, especially the intake of substances such as alcohol and nicotine due to their widespread use and accessibility. Studies indicate that smoking significantly increases the risk of GC. Non-drinking smokers experience an 80% increased risk of GC, and excessive alcohol consumption—both in smokers and non-smokers—is correlated with a statistically significant increase in the risk of GC [22]. In the European prospective nutrition cohort study, 444 cases of GCs were examined. Heavy alcohol intake was positively correlated with GC risk, whereas reduced alcohol intake was not correlated [23]. Based on a group from the Korean population showing the ALDH2 genotype, the relationship between alcohol intake and the risk of GC was examined. Among the group of patients who were ALDH2*1/2 carriers, those who were current drinkers or had a history of high alcohol intake were at a higher risk of cancer development compared to those who drank alcohol rarely or never. This study found a connection between alcohol intake and the development of GC among a group of individuals with ALDH2 polymorphisms and the ALDH2*1/*2 genotype [24]. The increased acetaldehyde levels induced by heavy alcohol drinking may lead to DNA damage and subsequently increase the risk of GC. Several studies have investigated the differential role of alcohol on GC risk according to polymorphisms in various genes (e.g., GSTT1, GSTM1, CYP2E1, SULT1A1, and ALDH2). It has been shown that high alcohol consumption may elevate GC risk more among individuals with low enzyme efficiency in their detoxification reactions [20,25].

In EOGC patients aged 20–39, several risk factors were compared with those observed in late-onset GCs based on the analysis of Surveillance, Epidemiology, and End Results (SEER) and Behavioral Risk Factor Surveillance System (BRFSS) data. These analyses showed that high alcohol consumption was positively correlated with EOGC, but obesity and smoking were not significantly correlated with EOGC [26].

The World Health Organization described H. pylori, a Gram-negative bacterium, as a class I carcinogen of GC development [27]. The carcinogenic mechanisms associated with *H. pylori* are based on chronic inflammation caused by infection with this bacterium and on bacterium-specific virulence factors that can damage the DNA of gastric epithelial cells and promote genome instability. The course of *H. pylori* infection is most often asymptomatic. Still, everyone will sooner or later develop gastritis, which in the long run may result in the appearance of stomach and duodenal ulcers and, ultimately, gastric cancer and lymphoma of the lymphatic tissue associated with the mucosa [28]. The virulence of the bacteria is primarily determined by the vacuolating cytotoxin (VacA) and the SON-related antigen (CagA) encoded by the DNA region located within the so-called pathogenicity islands (cag-PAI) [29,30]. VacA is considered a multifunctional toxin responsible for, among other things, inducing vacuolization, necrosis, and apoptosis of host cells. CagA can influence cell adhesion, spreading, and migration, as well as induce cytoskeletal rearrangements, influencing cell proliferation and stimulating gastric epithelial cells to secrete IL-8 [31]. Several potential mechanisms of carcinogenic action have been proposed, including DNA damage, activation of MAPK and B-catenin pathways, and p53 protein degradation. In addition to its direct mechanism of action, *H. pylori* also has an indirect effect, stimulating the carcinogenesis process by activating oxidative stress, initiating a chronic inflammatory process and modulating the immune response through factors such asSmad7 (suppressor of mothers against decapentaplegic 7), ROS (reactive oxygen species), RNS (reactive nitrogen species), IL-17, IL-21 (interleukin-17, interleukin-21), and NF-κB (nuclear factor kappa-light-chain-enhancer of activated B cells) [29,30,31]. The combination of specific polymorphisms of genes involved in the inflammatory response to *H. pylori* (IL-1 beta, TNF-alpha, and IL-10) and unfavorable bacterial genotypes (VacA and cag-PAI) may increase the risk of developing cancer by almost a hundredfold [29,30]. *H. pylori* infection has been classified as one of the risk factors leading to GC development by multiple epidemiological studies. *H. pylori* infection disrupts the microenvironment of the gastric tissue, leading to epithelial–mesenchymal transition (EMT) and further GC progression [2,32]. *H. pylori* infection plays a significant role as it contributes to the development of tumors in EOGC patients; however, no statistically significant difference in the distribution of IL-1 beta polymorphisms between young and old patients has been observed [16,33].

Another factor correlated with the development of GC is the Epstein–Barr virus (EBV). The rate of EBV infection in adults is 90%, but the incidence of gastric cancer remains low [34]. EBV was found to be present in only around 10% of GC cases; however, there is not enough evidence to form a distinct etiological role of EBV in GC development [35,36]. There are two theories regarding the mechanisms of EBV infection. First, EBV infects B lymphocytes and oral epithelial cells. EBV enters the gastrointestinal tract with saliva, directly infecting epithelial cells. The second theory is that the EBV virus is somehow reactivated in B lymphocytes in the stomach, and then, released to infect epithelial cells [34]. Gastric carcinomas that are EBV-positive vary depending on aspects such as the patient’s sex, age, or anatomic subsite. Studies show a diminishing tendency with age among men [37]. Cases of EOGC show a decrease or absence of EBV infections [38].

## 3. Classification System

In 1926, the Borrmann Classification System was proposed, based on macroscopic pathological evaluation or endoscopy after resection, and widely adopted for describing general or endoscopic lesions. According to this classification, advanced GC can be divided into four types based on macroscopic findings (Borrmann types I to IV) [39].

The most popular classification of GC, from 1965, is the Lauren classification, considering cell morphology and the infiltration method. The Lauren classification divides GC into two histological subtypes: intestinal and diffuse. Their different characteristics, including morphology, clinical features, and expansion properties, affect surgical decisions regarding the range of stomach resections and the long-term prognosis. A special type of diffuse subtype is GC with signet ring cells [40].

The World Health Organization (WHO) issued a classification of GCs in 2010, which is considered to be the most detailed among all classification systems and a reference point for describing gastric cancers. This WHO classification describes not only ubiquitous cancers such as stomach adenocarcinomas but also those types of gastric tumors with decreased occurrence [41]. Gastric adenocarcinomas can be divided into various subgroups, including tubular, mucinous, papillary, and mixed carcinomas, which, according to the Lauren classification system, are comparable to the indeterminate type. The signet ring cell carcinoma is an example of a dyscohesive carcinoma. All the other gastric adenocarcinomas, which do not belong to any category mentioned before, are described as uncommon, mainly because of their low clinical importance. According to the WHO classification, the most frequently occurring GC subtype is tubular adenocarcinoma, followed by the papillary and mucinous types. Around 10% of GCs are signet ring cell carcinomas, characterized by signet ring cells in over half of the tumor [42,43,44].

The development of GC may be related to genetic susceptibility factors. Single-nucleotide polymorphisms (SNPs), a common type of genetic mutation, may accelerate the development of GC. Genome-wide association studies (GWASs) may be able to identify sequence variations in the human genome, screen SNPs related to human diseases, and extend our understanding of the associations between genetic variations and cancer risk [45].

In a publication by Cristescu et al. [44], gene expression data of 300 primary gastric tumors were studied, which led to the molecular classification of GC based on NGS data. Four molecular subtypes of GC were selected, namely, MSS/TP53+, MSS/TP53−, MSI, and MSS/EMT subtypes.

High-throughput technologies now allow for a comprehensive study of genomic and epigenomic alterations associated with GC. Gene mutations, chromosomal aberrations, differential gene expression, and epigenetic alterations are some of the genetic/epigenetic influences on GC pathogenesis [46]. In 2014, The Cancer Genome Atlas Consortium (TCGA), based on key DNA defects and molecular abnormalities, divided GC into 8% Epstein–Barr virus (EBV)-positive, 22% microsatellite instability (MSI), 50% chromosomal instability (CIN), and 20% genomic stability (GS) types [47]. The TCGA typing is based on European and U.S. populations.

The occurrence of different modalities of GC development is shown in Figure 2.

The TNM is a specific type of GC classification used to determine the degree of progression to enable clinical and pathological comparisons. It includes the extent of the tumor (T), the extent of spread to the lymph nodes (N), and the presence of metastasis (M). The T category describes the original (primary) tumor. The M category tells whether there are distant metastases (the spread of cancer to other parts of the body).

## 4. Genetic and Molecular Alteration in GC Development

There are many articles describing genetic polymorphisms regarding the risk of GC. The analysis of multiple studies on molecular biomarkers of GC established a wide spectrum of identification patterns in this field. However, the research also indicated that risk factors for conventional GC are less correlated with EOGC [48].

**HER2**—A proto-oncogene expressed in 15–37% of gastric adenocarcinomas and an ideal target for inhibition in malignancy with high recurrence and dismal survival rates. HER2/neu amplification is higher in the intestinal histologic subtype of GC and is not associated with gender and age; however, it is associated with poor survival [49,50]. The ToGA trial, which added trastuzumab (monoclonal antibody) to standard chemotherapy, showed improved survival of patients with HER2-positive advanced G/GEJ adenocarcinoma and brought these patients into a new era of HER2-targeted therapy. EOGCs show different molecular profiles than GCs in older patients (>45 years). This difference is emphasized by the studies by Moelans et al., who showed that EOGC has a lower frequency of HER2 amplification and overexpression than conventional GC [51].

**VEGF** (vascular endothelial growth factor)—Performs its function mainly through binding to vascular endothelial growth factor receptor-2 (VEGFR2) via auto-phosphorylation mechanisms that activate downstream signal pathways involved in endothelial cell (EC) proliferation, survival, and motility, playing a key role in physiological and pathological angiogenesis [52,53]. VEGF plays a crucial role in tumor growth, even in an angiogenesis-independent way, by interacting with receptors expressed on tumor cells through autocrine and/or paracrine mechanisms [54]. Similarly to other neoplastic diseases, the treatment of GC may benefit from anti-angiogenic drugs, including ramucirumab, a monoclonal antibody antagonist of VEGFR2, currently used in the second-line therapy of GC and gastro-esophageal junction carcinomas. Based on the results of two different randomized phase III trials, ramucirumab could be used as monotherapy or in combination with paclitaxel (PTX) in pre-treated patients after platinum- and fluoropyrimidine-based therapy [40].

**Microsatellite instability (MSI)**—A critical marker for DNA mismatch repair deficiency, contributing to an increased accumulation of genetic alterations in gastric carcinogenesis. MSI-positive patients exhibit a low frequency of specific mutations, including those identified in the PIK3CA, EGFR, ERBB3, and ERBB2 genes. Gastric cancer cases characterized by high MSI levels may confer long-term survival benefits, irrespective of positive resection margin status [23,55]. In a pooled meta-analysis of four randomized controlled trials of resected GC, high MSI status was associated with longer overall survival (OS) and a lack of benefit from perioperative or adjuvant chemotherapy [56]. In the study, Bergquist et al. indicated that EOGC was more likely to be a genomically stable subtype (22.5% vs. 8.1%). In contrast, late-onset GC was more likely to be a microsatellite instability subtype (18.6% vs. 5.6%) [48].

**PDL1**—Patients without any EBV-positive metastasis with PCNA and C-met-expression show elevated levels of PDL1; studies show a more prospective result when PD-L1/PD-1 expression is elevated [57]. The increased expression of PD-L1 in GC is related to the epithelial–mesenchymal transformation phenotype (EMT), which can further increase the potential of tumor metastasis. Immune checkpoint inhibitors, in combination with standard therapy, improve OS in both HER2-negative and HER2-positive GC according to the KEYNOTE-811, CheckMate-649, and KEYNOTE-859 trials [58,59,60].

**CDH1**—Germline alterantions of the tumor suppressor gene (E-cadherin) are associated with hereditary diffuse gastric cancer (HDGC) and occur in approximately 40% of HDGC families. The E-cadherin protein is essential for cell proliferation, the maintenance of cell adhesion, cell polarity, and epithelial–mesenchymal transition. Dysregulation leads to tumor proliferation, invasion, migration, and metastasis. Germline mutations in CDH1 have been documented in several patients with early-onset diffuse gastric cancer (EODGC) without a family history, but the true incidence in this case is unknown. The advanced stage of diagnosis remains a clinical burden for these patients due to poor long-term survival [16,61]. CDH1 is considered a predisposing gene alongside CTNNA1 (CTNNA1 encodes for α-E-catenin, a partner of E-cadherin in the adherens junction complex). As with CDH1-related disease, HDGC families diagnosed with CTNNA1 and harboring truncating variants are advised to undergo annual endoscopy screening with multiple random biopsies (Cambridge protocol) and, in the case of positive biopsies, to undergo prophylactic total gastrectomy [62].

**CCND1**—Cyclin D1 functions as a positive regulator of the cell cycle, while retinoblastoma protein (pRb) acts as a repressor by promoting G1/S arrest and growth restriction by inhibiting E2F transcription factors. The elevated expression of both cyclin D1 and pRb is associated with cell overgrowth and cancer development. The expression of pRb and cyclin D1 may be evident in the early stages of gastric carcinogenesis, with higher expression levels observed in non-neoplastic mucosa conditions such as dysplasia, intestinal metaplasia, atrophy, and gastritis, progressing to carcinoma [63].

**p53 gene**—Mutations in the p53 gene occur in the early stages of gastric cancer (GC) and become more frequent in advanced stages of cancer progression. Patients with TP53-positive tumors are categorized as a distinct subtype of GC [64].

**Bcl-2**—The negative expression of Bcl-2 is associated with an increased chance of cancer recurrence, lymph node metastases, and depth of invasion [65].

**Mucins**—A class of extracellular, high-molecular-weight, heavily glycosylated proteins that play critical roles in cell signaling, the formation of chemical barriers, gel formation, and lubrication. Additionally, they exhibit significant inhibitory functions. Elevated expression levels of mucin proteins, including MUC1, MUC2, MUC5AC, and MUC6, are implicated in the process of gastric carcinogenesis [66,67].

**MRP2**—The overexpression of MRP2 is notable for its association with the initial lack of response to chemotherapy treatments in tumors, positioning it as a significant biomarker for predicting chemotherapy response [68].

**GST-P**—The expression of GST-P is markedly elevated in chemically induced tumors. It is also correlated with tumor invasion and recurrence, as well as poor prognosis [69].

In 2019, Tian et al. conducted a meta-analysis to summarize and assess the credibility and strength of genetic polymorphisms on the risk of GC, mainly in the Asian population. In summary, the study found nine variants in nine genes, which were rated as demonstrating strong evidence of association with GC risk, including APE1 rs1760944, DNMT1 rs16999593, ERCC5 rs751402, CASP8 rs3834129, GSTT1 null/presence, MDM2 rs2278744, PPARG rs1801282, TLR4 rs4986790, and IL-17F rs763780 [11].

***APE1***—Apurinic/apyrimidinic endonuclease 1, located on chromosome 14q11.2, participates in DNA base excision repair and has been associated with human carcinogenesis. This analysis provides strong evidence for a connection between the G allele of the APE1 polymorphism and GC risk [11].

***DNMT1***—Located on human chromosome 19p13.2, DNMT1 encodes a protein comprising 1.632 amino acids, possibly associated with carcinoma development. Some studies have suggested that DNA methylation contributes to the progression of GC and that overexpression of DNMT1 may be associated with GC risk [11].

***ERCC5***—Also known as XPG, ERCC5 is an endonuclease that may prevent carcinogenesis by excising damaged DNA during DNA repair. A polymorphism (rs751402) is found in the promoter region of ERCC5 and controls its expression and function during transcription in healthy human cells. A study showed that this SNP in a dominant model was strongly associated with an increased risk of GC [11].

**Human *CASP8***—Located on chromosome 2q33–q34, CASP8 participates in cell cycle regulation. An SNP (rs3834129) located in the promoter region of CASP8 leads to the reduced expression of this gene. Impaired CASP8 expression can decrease T lymphocyte-induced cell death. In the additive model, this SNP was strongly associated with GC. This polymorphism could present a novel target for gene therapy of GC and lead to new drug developments against GC [11].

In a European study, Machlowska et al. compared and characterized age-dependent genotypic and phenotypic characteristics of GC subtypes using high-throughput sequencing of GC patients [70]. The results allowed the authors to identify potential genes distinguishing between EOGC and CGC. Within the two subgroups studied, seven candidate genes and nine variants were identified, with the statistical significance of these genes varying between subgroups. Variants such as rs1799939 (RET), rs2959656 (MEN1), and rs55986963 (KIT) were predominantly observed in EOGC patients (Table 1). Notably, rs2959656 was homozygous in 100% of EOGC cases. Variant rs1799939 in the RET gene is associated with multiple endocrine neoplasia, hereditary cancer-predisposing syndromes, renal dysplasia, and pheochromocytoma, as reported in the NCBI ClinVar database. Activation of the RET proto-oncogene is believed to drive gastric inflammatory and neoplastic diseases. Variant rs2959656 is found in patients with hereditary endocrine cancer syndrome multiple endocrine neoplasia type 1 (MEN1). At the same time, rs55986963 in the KIT gene is closely linked to gastrointestinal stromal tumors (GISTs), mastocytosis, and partial albinism [70].

South Korea has one of the highest rates of EOGC globally, with 15% of GC cases diagnosed in individuals under 45 years old [71]. Korean researchers analyzed the EOGC protein genome in 80 patients, revealing distinct gene expression profiles compared to conventional GC. They identified six significantly mutated genes: CDH1, TP53, BANP protein, mucin 5B, transforming protein RhoA, and AT-rich interactive domain-containing protein 1A (Table 1). However, no differences in mRNA expression patterns were detected between EOGC and late-onset GC [72].

Integrating biomarker testing, especially the analysis of HER2 status, microsatellite instability (MSI) status, and programmed death-ligand 1 (PD-L1) expression, has revolutionized clinical practices and patient care. Targeted therapies like trastuzumab, nivolumab, and pembrolizumab have shown promising results in clinical trials for treating locally advanced or metastatic disease. Palliative management, including systemic therapy, chemoradiation, and best supportive care, is recommended for all patients with unresectable or metastatic cancer [73].

Understanding the genetic signatures underlying GC development is crucial for advancing personalized medicine and identifying future treatment strategies.
cancers-16-03163-t001_Table 1Table 1Mutated variants and genes in patients with EOGC.Origin of the Study GroupNumber of PatientsMutated Variants/GenesAuthorsThe Netherlands, Finland, and Poland35 patients with EOGCrs1799939 (RET);rs2959656 (MEN1);rs55986963 (KIT).Machlowska et al. [70]South Korea80 patients with EOGCCDH1;TP53;BANP protein;mucin 5B;transforming protein RhoA;AT-rich interactive domain-containing protein 1A.Mun et al. [72]

## 5. Probable Biomarkers of Gastric Cancer

Carbohydrate antigen CA 19-9 (CA19-9) is the most commonly used serum tumor marker for GC. However, it may also be overexpressed in pancreatic cancer and in other gastro-intestinal tumors [74]. In GC, CA19-9 is mainly useful for checking for relapse and monitoring metastatic disease. The usefulness of CA19-9 as a diagnostic biomarker of GC remains slightly controversial because the results of the studies in which it was used were usually contradictory. However, the tumor depth, tumor stage, and lymph node metastasis in GC patients are characteristics that may be associated with CA19-9 [75,76,77]. Increased concentrations of CA19-9 can also constitute a marker for early recurrence after therapeutic gastrectomy for GC, and they can also be an indicator of potential peritoneal dissemination and increased mortality [78,79,80]. In their study, Song et al. reported that higher CA19-9 levels are most commonly found in cases of the stage III/IV group GC relative to the I/II group [81]. To take any further diagnostic steps based on biomarker levels, the combined detection of several markers seems to be inherent. Studies show that in GC screening, the combination of CEA and CA19-9 in serum achieves higher specificity than CEA in serum alone, and the combination of CEA, CA125, and CA19-9 has been shown to provide higher sensitivity than CEA alone [75,82]. In early GC diagnosis and therapy monitoring, serum CA19-9, carcinoembryonic antigen (CEA), and carbohydrate antigen 72-4 (CA72-4) are the most important biomarkers used [83,84].

## 6. Treatment Strategies

### 6.1. Surgery

Surgery plays a crucial role in treating GC. Gastrectomy with D2 lymph node dissection is the main surgical procedure with curative intent. In tumors localized in the mid-distal part of the stomach, distal gastrectomy preserving the upper third of the stomach is considered equivalent to total gastrectomy, which is reserved for proximal tumors. The preferable proximal margin of resection for peripheral subtotal gastrectomy should be 5 cm for tumors T2 or deeper with an infiltration growth pattern and 3 cm for tumors with an expansive growth pattern. There are no distinctions for histological types of cancer. For tumors invading the esophagus, R0 resection should be ensured, with a frozen section examination of the resection line. In patients with advanced gastric cancer (>cT1b) or with cN+, with the intention of radical treatment, D2 lymphadenectomy is routinely recommended. D1/D1+ lymphadenectomy is allowed in patients operated on for early cT1a cancers that do not meet the criteria for EMR/ESD and for cT1bN0 tumors that show a histopathologically high degree of differentiation and are 1.5 cm or smaller in diameter. Routine splenectomy is not recommended, but it should be performed if the spleen or its hilar lymph nodes are invaded. Omentectomy is integrated into standard gastrectomy for T3 or deeper tumors. For T1/T2 tumors, the omentum more than 3 cm away from the gastroepiploic artery can be preserved. Extended surgery with multi-organ resection can be considered in order to achieve the oncological radicality of the procedure (R0) [85].

### 6.2. Palliative Surgery

Palliative, non-radical interventions may be used in particular circumstances. Gastric resection is recommended to alleviate complications associated with GC (bleeding, obstruction, and perforation). Performing resection procedures with the intention of cytoreduction is not recommended in patients with no indications for palliative resections to alleviate complications related to the tumor (bleeding, obstruction, and perforation). In the case of palliative resection, lymphadenectomy beyond the scope of D1 is not recommended. In patients with isolated distant metastases, R0 resection of the primary tumor and metastases is allowed [85].

### 6.3. Minimally Invasive Surgery

The role of minimally invasive surgery, such as laparoscopy and robotic-assisted surgery, is growing. In patients with early gastric cancer, laparoscopic distal gastrectomy is considered equivalent to laparotomy in centers with appropriate experience. However, for early cancer, an equivalent procedure, such as laparoscopic total gastrectomy, is also considered [85].

Endoscopic procedures are allowed in selected patients with EOGC with the intention of healing as well as in tumors with a low possibility of lymph node metastasis and suitable for en bloc resection. Because endoscopic resection only involves local treatment without the removal of lymph nodes, as a rule, it is performed only when lymph node metastases do not occur [86,87]. Endoscopic mucosal resection (EMR) procedures and submucosal dissection (ESD) for GC should only be performed in centers with appropriate experience in this area. It was agreed that the basic indication for performing EMR procedures in patients with EOGC are lesions that meet the following criteria: a high degree of histological differentiation (G1), no ulceration (UL0), invasion limited to the mucosa (cT1a), and a lesion diameter up to 2 cm.

In turn, it was agreed that the basic indications for performing ESD procedures in patients with EOGC are lesions that meet the following criteria: a high degree of histological differentiation (G1), no ulceration (UL0), infiltration limited to the mucosa (cT1a), and a lesion diameter greater than 2 cm [88].

The completeness of the primary tumor removal after EMR/ESD must be evaluated according to the JGCA criteria for endoscopic curability, which are different for tumors with a dominant diffuse or undifferentiated type [89]. Regardless of whether the radicality of resection is certain, particularly when the lesion is not resected en bloc or has a positive horizontal/vertical margin, surgical resection of the stomach should be considered.In the case of cancer recurrence limited to the mucous membrane after endoscopic surgery performed in accordance with basic indications, it is permissible to repeat the procedure once using the submucosal dissection technique [90,91,92].

## 7. Neo-Adjuvant and Adjuvant Chemotherapy and Radiotherapy

Currently, the treatment of conventional GC, as well as EOGC, is mainly based on surgery combined with chemotherapy/targeted therapy and immunotherapy [93]. In every patient with advanced GC (>cT1b), indications for combined treatment within a multidisciplinary team should be considered. Any patient with potentially resectable GC, cT2 stage, any N, M0 for whom R0 surgery is assumed, and in whom there are no indications for urgent gastrectomy should be qualified for perioperative chemotherapy.

Based on the results of clinical trials, the recommended standard pre-operative treatment regimen is FLOT chemotherapy (5-FU-leucovorin-oxaliplatin-docetaxel). This combination therapy consists of a pre-operative four-cycle chemotherapy and a post-operative four-cycle chemotherapy [94]. For patients unfit for triplet chemotherapy, a combination of fluoropyrimidine with oxaliplatin (FOLFOX/XELOX) is recommended. In patients undergoing perioperative chemotherapy, the use of post-operative radiotherapy does not provide additional benefits. In patients who receive adequate surgery and have a high risk of relapse (e.g., positive nodal status), only adjuvant chemotherapy should be given. In patients with stage IB or higher who did not receive chemotherapy before resection, adjuvant chemotherapy is recommended; in these patients, adjuvant radiotherapy may be used if they did not receive D2 lymphadenectomy [95,96]. No specific and evidence-based recommendation can be made for patients with R1-resected GC [97].

In patients with advanced, locally unresectable tumors but without distant metastases (T4b, any N, and M0), the use of induction chemotherapy should be considered, followed by gastrectomy if the tumor becomes operable. In patients with advanced, unresectable, or metastatic GC, chemotherapy regimens are recommended, involving a combination of two or three drugs, including a platinum derivative and fluoropyrimidine [98]. In patients with advanced, unresectable, HER2-expressing GC, systemic therapy is recommended, including trastuzumab in combination with a platinum derivative and fluoropyrimidine [99].

Peritoneal diffusion is common in GC and produces a very short life expectancy (3–6 months). In limited peritoneal carcinomatosis, hyperthermic intraperitoneal chemotherapy (HIPEC) has been used. In a skilled team, the treatment is safe, but its efficacy is limited and should be confirmed [100,101,102,103]. Depending on the localization of carcinomatosis, peritonectomy may be a therapeutic option able to confer a survival benefit in selected patients, alone or in combination with early post-operative intraperitoneal chemotherapy (EPIC) [104,105]. In patients with unresectable peritoneal disease, it is possible to use the recently developed pressurized intraperitoneal aerosol ChT (PIPAC) technique, which enables homogeneous, locoregional administration of intraperitoneal ChT during a laparoscopic procedure [106]. Zhang et al. analyzed 1639 patients with EOGC (<50 years) and showed that patients who received surgical treatment had a better prognosis than those who received SROC, SCRT, or non-surgical treatment. Additionally, it was found that the highest risk of death occurred in patients who received non-surgical treatment. The authors also showed that additional radiotherapy, chemotherapy, or chemoradiotherapy did not provide a coordinated survival benefit in patients with EOGC [106].

### Preventive Measures in Gastric Cancer Development

Numerous studies have demonstrated that dietary and lifestyle factors can play a preventive role in gastric cancer. These factors include consuming fruits and vegetables rich in vitamins C, A, and E; adherence to a Mediterranean diet; intake of dairy products, cruciferous vegetables, and dietary fiber; and regular physical activity. On the other hand, excessive consumption of coffee, salt, and saturated fats in the diet; intake of red meat; and the presence of nitrates, nitrites, and nitrosamines in food products are unequivocally considered risk factors for developing gastrointestinal cancers [107,108].

Recent research highlights the significant preventive role of polyphenols in developing gastric cancer, as they may inhibit cancer cell growth. Preliminary evidence from epidemiological studies suggests that a diet regularly including fruits and vegetables (rich in polyphenols) significantly reduces the risk of carcinogenesis [109].

Studies also indicate the protective role of flavonoids in gastric cancer development. Research by Storelli et al. (2019) demonstrated that consuming flavonoid-rich products can act as a protective factor, reducing the risk of gastric cancer by 24% to 40% [109]. The anticancer properties of polyphenols are largely attributed to their strong anti-inflammatory and antioxidant potential, as well as their ability to modulate cellular signaling pathways and molecular targets [110,111].

Flavonoids are secondary metabolites present in many plants, involved in photosynthesis and the defense against pathogens and ultraviolet radiation. The human body cannot synthesize flavonoids; thus, they must be supplied through plant-based products. Consuming flavonoids can reduce the risk of cancer development through various mechanisms, such as protecting against DNA damage, blocking cellular pathways leading to cancer, inducing apoptosis, modulating cell proliferation, inhibiting angiogenesis, and preventing invasive tumor growth. Additionally, certain flavonoids exhibit antimicrobial activity against *Helicobacter pylori* [110].

Bioactive compounds found in numerous plants and herbal mixtures, such as resveratrol, cinnamaldehyde, oleuropein, shikonin, and magnolol, have also shown significant anticancer activity against GC. Studies on the molecular docking of these herbal components with key genes have indicated that TLR4 can bind closely with salvianolic acid B, suggesting that salvianolic acid B has strong efficacy in the treatment of gastric cancer both in vitro and in vivo, as TLR3 and TLR4 are highly expressed in cancer cells [112]. Thus, the anticancer mechanisms mainly include the induction of apoptosis, autophagy, and cell cycle arrest. Therefore, herbs and herbal components hold great potential for improving the prognosis of GC patients by targeting TLRs, as well as providing alternative options for preventive measures and protective actions against gastric cancer development [113].

Another currently widely researched approach to cancer treatment is combining phytotherapy with conventional anticancer therapies to inhibit the process of angiogenesis [113].

## 8. Conclusions

This review aims to raise awareness of GC, including EOGC, based on available scientific reports. There are fewdata on the epidemiology, histopathology, risk factors, and genome characteristics of EOGC. The observed increase in morbidity and mortality due to this disease demonstrates the need to better define these features based on the currently available literature and experimental studies.

## Figures and Tables

**Figure 1 cancers-16-03163-f001:**
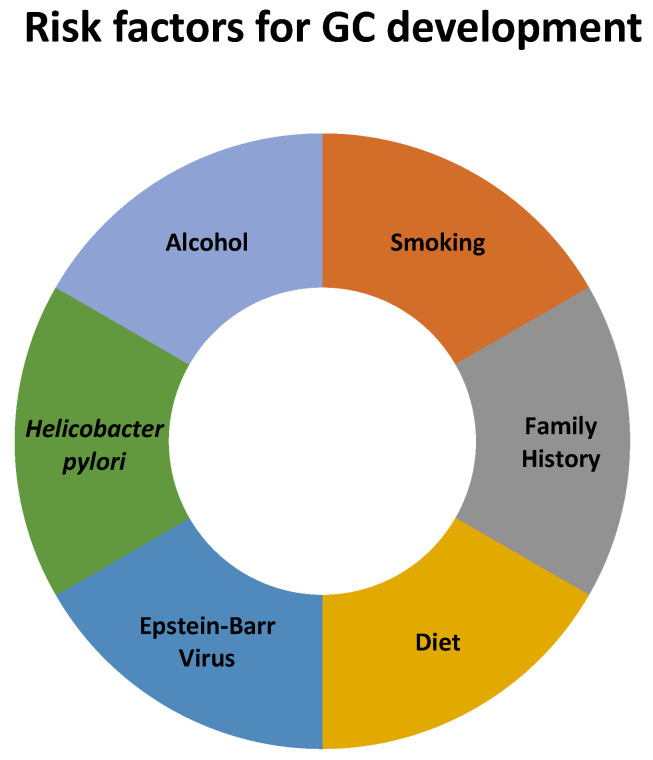
The risk factors for GC development.

**Figure 2 cancers-16-03163-f002:**
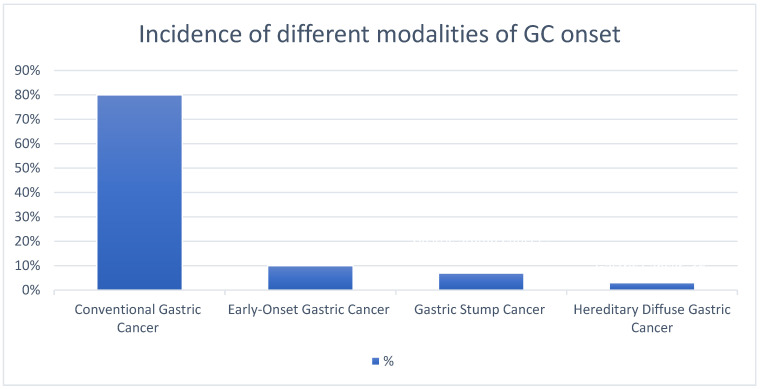
The different modalities of GC onset.

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
