# Peer review of "Gastric Cancer: An Up-to-Date Review with New Insights into Early-Onset Gastric Cancer"

_cancers, 2024, doi:10.3390/cancers16183163_

Round 1
Reviewer 1 Report
Comments and Suggestions for Authors
The title of manuscript is rather misleading. Indeed, the review mainly consider gastric cancer and only few hints are reserved to early-onset gastric cancer (EOGC). As AA themselves admit in the “Conclusion”, biological and etiopathogenetic data specifically regarding EOGC are very scarce or not existing. So, the title should refer only to gastric cancer. To extrapolate the few observations on EOGC, a table should be useful, summarizing the limited knowledge on this entity: epidemiologic, anatomic, histologic, genetic and surgical aspects as a consequence of its localization.
“Abstract” is for the most part a repeat of “Introduction”.
Many references do not match with sentences to which they are referred. It is necessary to check them and eliminate inconsistencies.
Line 117. Nicotine and alcohol are better definite as voluptuary than psychoactive substances.
Lines 121-122. It is not clear what AA mean with the sentence “insignificant increase in risk of 80%”.
Lines 142-170. The reasoning on the role of EBV and HP should be better and more clearly developed, as well the role of bacterial overgrowth. Note that EBV genome is not present in all gastric cancer, but only in a sub-group.
Lines 181-210. Classification systems of gastric cancer evolved over time, with TGCA molecular classification being the most recent and, probably, the most inclusive among those discussed. So, a chronological order might better represent the evolution of knowledge in the last 20 years. Furthermore, also the Bormann classification should be mentioned in the appropriate place.
Lines 211-212. Figure 2 should be better introduced. Differently from classification systems, this classification does not describe “subtypes”, but different modalities of onset; therefore, it seems distinct from the classifications previously mentioned. The heading used for table 2 (“Gastric cancer subtypes”) sounds inappropriate.
Lines 215-218. AA should explain that TNM is a particular type of classification. It does not refer to a pathogenetic mechanism and/or clinical behavior of gastric cancer. TNM is a staging system used for describing the level of progression for the purpose of allowing clinical and pathological comparisons.
Paragraph 5. Not all the genetic alterations in gastric cancer are polymorphisms. Therefore “Genetic and molecular alteration” seems a more appropriate heading for this paragraph. Some concepts here reported are obscure and difficult to understand due to not clear English language. The biomarkers described in this section have been usually studied as prognostic or predictive factors, rarely as risk factors. For most of the molecular alterations here reported it is not specified the role in early -onset gastric cancer. If AA assume that EOGC has a particular genetic framework, this must result in their review, otherwise they are just describing the genetic alterations in gastric cancer. In the last years several reviews have been devoted to genetic alterations in gastric cancer. CDH1 gene mutation is one of the few germ-line mutations responsible for hereditary gastric cancer in association with breast cancer, hereditary diffuse gastric cancer (HDGC); especially in the context of EOGC, it deserves an accurate description as well its relationship with CTNNA1.
The general arrangement of this paragraph should include just a brief overview of genetic alterations in gastric cancer and, if it would be possible, a more detailed description of those alterations for which some difference has been found in EOGC.
Lines 247-249. Clarify how increased mutational load in MSI-H produces a “low frequency of specific mutations”. AA probably refer to mutations in some target gene.
Line 262. I do not know what kind of drug Ramoximab is; it is cited also in Table 1. The search in PubMed and with different search engines (Google, Bing, Yahoo) does not retrieve any result for Ramoximab.
Line 264. This is one of the statements that is not consistent with the reference, as previously indicated; reference #64 seems more appropriate for the following paragraph (CCND1). Please, check carefully the references and quote them in the correct position.
Table I is very redundant and largely repeats what is described in the text. References are not consistent. In the present form the table is useless and makes heavy the manuscript.
Lines 314-334. These paragraphs address specifically molecular characteristics of EOGC and deserve a deeper attention. A table or a figure depicting such alterations might be useful.
Lines 350-352. This concept is included in the following statement (lines 352-353).
Lines 358-360. Please, describe the concept more clearly.
Line 364 and following lines. Surgery may be a sufficient heading. AA have to specify that tumor site, proximal or distal, is the major determinant in the choice of total or sub-total gastrectomy, that should be always combined with D2 lymphadenectomy. Moreover, a more ordered description of surgical options is advisable, also including palliative interventions.
Lines 386-389. These statements are confusing. Start a new section for palliative surgery and describe clearly when palliative resection or other palliative surgical procedures are indicated.
Line 392. Omentectomy may also be part of radical surgery and should be reported before describing palliative procedures.
Line 394. Start a new section for minimally invasive surgery.
401-403. Indication for endoscopic procedures might be better described.
Line 410. Are AA sure that “changes” is the proper term?
Line 448. In limited peritoneal carcinomatosis, also peritonectomy should be considered, not only HIPEC/PIPAC, because it is more effective in combination with systemic chemotherapy or HIPEC.
Lines 460-461 This statement seems rather inconsistent with the previous lines 457-458.
Comments on the Quality of English LanguageEnglish Language must be extensively revised, including punctuation. Some phrases are obscure and difficult to understand.
Author Response
Dear Madam/Sir,
Thank you for the favourable review and comments We did our best to improve our manuscript. Below we enclose brief answers to yours comments:
The title of manuscript is rather misleading. Indeed, the review mainly consider gastric cancer and only few hints are reserved to early-onset gastric cancer (EOGC). As AA themselves admit in the “Conclusion”, biological and etiopathogenetic data specifically regarding EOGC are very scarce or not existing. So, the title should refer only to gastric cancer. To extrapolate the few observations on EOGC, a table should be useful, summarizing the limited knowledge on this entity: epidemiologic, anatomic, histologic, genetic and surgical aspects as a consequence of its localization.
-As we indicated in the article, there are a limited number of reports regarding EOGC, therefore it is difficult to provide more information on this topic in the review. However, as authors, we make every effort to also refer to information regarding EOGC for each aspect of GC discussed. Therefore, it seems reasonable for us to include information in the title that the article will not only concern conventional GC.
“Abstract” is for the most part a repeat of “Introduction”.
- “Abstract” has been updated with additional information.
Many references do not match with sentences to which they are referred. It is necessary to check them and eliminate inconsistencies.
-We checked and corrected it.
Line 117. Nicotine and alcohol are better definite as voluptuary than psychoactive substances.
-We checked and corrected it.
Lines 121-122. It is not clear what AA mean with the sentence “insignificant increase in risk of 80%”.
-The sentence has been removed from the text due to maintain clarity of the manuscript.
Lines 142-170. The reasoning on the role of EBV and HP should be better and more clearly developed, as well the role of bacterial overgrowth. Note that EBV genome is not present in all gastric cancer, but only in a sub-group.
-We described it in more detail.
Lines 181-210. Classification systems of gastric cancer evolved over time, with TGCA molecular classification being the most recent and, probably, the most inclusive among those discussed. So, a chronological order might better represent the evolution of knowledge in the last 20 years. Furthermore, also the Bormann classification should be mentioned in the appropriate place.
-We changed it in accordance with the reviewer's recommendation.
Lines 211-212. Figure 2 should be better introduced. Differently from classification systems, this classification does not describe “subtypes”, but different modalities of onset; therefore, it seems distinct from the classifications previously mentioned. The heading used for table 2 (“Gastric cancer subtypes”) sounds inappropriate.
-We changed the heading used for table 2.
Lines 215-218. AA should explain that TNM is a particular type of classification. It does not refer to a pathogenetic mechanism and/or clinical behavior of gastric cancer. TNM is a staging system used for describing the level of progression for the purpose of allowing clinical and pathological comparisons.
-We corrected it.
Paragraph 5. Not all the genetic alterations in gastric cancer are polymorphisms. Therefore “Genetic and molecular alteration” seems a more appropriate heading for this paragraph. Some concepts here reported are obscure and difficult to understand due to not clear English language. The biomarkers described in this section have been usually studied as prognostic or predictive factors, rarely as risk factors. For most of the molecular alterations here reported it is not specified the role in early -onset gastric cancer. If AA assume that EOGC has a particular genetic framework, this must result in their review, otherwise they are just describing the genetic alterations in gastric cancer. In the last years several reviews have been devoted to genetic alterations in gastric cancer. CDH1 gene mutation is one of the few germ-line mutations responsible for hereditary gastric cancer in association with breast cancer, hereditary diffuse gastric cancer (HDGC); especially in the context of EOGC, it deserves an accurate description as well its relationship with CTNNA1. The general arrangement of this paragraph should include just a brief overview of genetic alterations in gastric cancer and, if it would be possible, a more detailed description of those alterations for which some difference has been found in EOGC.
-We changed it in accordance with the reviewer's recommendation.
Lines 247-249. Clarify how increased mutational load in MSI-H produces a “low frequency of specific mutations”. AA probably refer to mutations in some target gene.
-We indicated the mutations in line 250.
Line 262. I do not know what kind of drug Ramoximab is; it is cited also in Table 1. The search in PubMed and with different search engines (Google, Bing, Yahoo) does not retrieve any result for Ramoximab.
-It must be our mistake. We removed this information.
Line 264. This is one of the statements that is not consistent with the reference, as previously indicated; reference #64 seems more appropriate for the following paragraph (CCND1). Please, check carefully the references and quote them in the correct position.
-We checked and corrected it.
Table I is very redundant and largely repeats what is described in the text. References are not consistent. In the present form the table is useless and makes heavy the manuscript.
-The table was removed.
Lines 314-334. These paragraphs address specifically molecular characteristics of EOGC and deserve a deeper attention. A table or a figure depicting such alterations might be useful.
-We added the table.
Lines 350-352. This concept is included in the following statement (lines 352-353).
-We modified it.
Lines 358-360. Please, describe the concept more clearly.
-We described the concept in more detail.
Line 364 and following lines. Surgery may be a sufficient heading. AA have to specify that tumor site, proximal or distal, is the major determinant in the choice of total or sub-total gastrectomy, that should be always combined with D2 lymphadenectomy. Moreover, a more ordered description of surgical options is advisable, also including palliative interventions
Lines 386-389. These statements are confusing. Start a new section for palliative surgery and describe clearly when palliative resection or other palliative surgical procedures are indicated.
-We changed it.
Line 392. Omentectomy may also be part of radical surgery and should be reported before describing palliative procedures.
-We changed it as indicated.
Line 394. Start a new section for minimally invasive surgery.
-We did it.
401-403. Indication for endoscopic procedures might be better described.
-Detailed indications for endoscopic procedures are described in lines 405-408.
Line 410. Are AA sure that “changes” is the proper term?
-We changed it to “lesions”.
Line 448. In limited peritoneal carcinomatosis, also peritonectomy should be considered, not only HIPEC/PIPAC, because it is more effective in combination with systemic chemotherapy or HIPEC.
-We added these information.
Lines 460-461 This statement seems rather inconsistent with the previous lines 457-458.
-We removed this sentence at this place and placed it between lines 420 and 421.
Comments on the Quality of English Language
English Language must be extensively revised, including punctuation. Some phrases are obscure and difficult to understand.
-The manuscript was carefully checked and revised by all authors as well as by an English native speaker.
Reviewer 2 Report
Comments and Suggestions for Authors
The authors deal with a review associated with gastric cancer and the new insights into 2 early-onset gastric cancer. The article corresponds to the title. The whole set up of the study is well organized. The abstract is informative. It is an informative review which will help readers to understand basic aspects of gastric cancer epidemiology, molecular profile and basics of treatment. As far as it concerns surgical treatment the authors provide a general and targeted approach. For tumors invading esophagus or in gastroesophageal junction IVOR LEWIS is the appropiate surgical treatment option. The peripheral gastrectomy even is suitable for peripheral tumors we prefer the subtotal gastrectomy for better oncological outcomes. The laparoscopic approach depends on the stage of the cancer , the site and the experience of the surgeon. The references are adequate and up to date. It is a well written review providing important knowledge about gastric cancer.
Author Response
Dear Madam/Sir,
Thank you for the favorable review and comments We did our best to improve our manuscript. The improved manuscript in the attachment.
Round 2
Reviewer 1 Report
Comments and Suggestions for Authors
I appreciate AA. efforts to comply with previous suggestions, but a number of criticisms remains and, in my opinion, some roughness and convolution worsens understanding and fluency of the manuscript.
Here, I report some further suggestions.
- Heading of figure 2. Substitute "onsets" with "onset"
- Line 313. Add "Immune checkpoint inhibitors in combination with standard therapy improve OS in both HER2 negative and HER2 positive GC according with KEYNOTE-811, CheckMate-649 and KEYNOTE-859 trials" (ref. Janjigian YY Lancet 2023 and JCO 2024; Sun Young Kha Lancet)
- Line 435. I am not sure that Ca 15.3 may be used as a biomarker in GC
- Lines 415-418. I suggest to reverse the statements: "Carbohydrate antigen Ca 19.9 (Ca 19.9) is the most commonly used serum tumour marker for GC. However, it may be overexpressed also in pancreatic cancer as well as in other gastro-intestinal tumours (74). In GC Ca 19.9 is mainly useful in checking for relapse and monitoring metastatic disease. The usefulness ..."
- Line 439. Delete "as a strategy"
- Lines 439-445. Substitute with: "Gastrectomy with D2 lymph node dissection is the main surgical procedure with intention of cure. In tumours localized in the mid-distal part of stomach, distal gastrectomy preserving the upper third of stomach is considered equivalent to total gastrectomy, which is reserved to proximal tumours. Preferable proximal margin of resection ..."
- Lines 448-449. Delete or explain better the sentence "In the case of tumours located in the proximal part of the stomach, upper resection is possible"
- Line 450. Substitute "it is preferable to ensure R0 resection" with "R0 resection should be ensured"
- Line 455. Pospone this sentence at the end of the paragraph
- Lines 456-458. Substitute with "Routine splenectomy is not recommended but it should be performed if spleen or its hilar lymph nodes are invaded"
Line 463. Substitute with "Palliative, non-radical, interventions may be used in particular circumstances. Gastric..."
- Line 464. Substitute "a cancerous tumour" with "GC"
- Line 467. Delete "cancer"
- Lines 469-470. Substitute "...it is allowed to resect the primary tumour along with metastases if it is possible to achieve R0" with "R0 resection of the primary tumour along with metastases is allowed"
- Lines 475-476. Delete "which is emphasized by many randomized controlled trials in progress"
- Line 476. Substitute "peripheral surgery such as laparoscopic resection" with "laparoscopic distal gastrectomy is considered equivalent to laparatomy..."
- Lines 494-498 The clinical sense of the sentences is unclear without a larger explanations. In the context of the present review I think better substitute them with "Completeness of the primary tumour removal after EMR/ESD must be evaluated according to JGCA criteria for endoscopic curability, that are different for tumours with dominant diffuse or undifferentiated type (ref. https://doi.org/10.1007/s10120-020-01042-y). Whether radicality of resection is not sure, in particular when the lesion is not resected en-bloc or had a positive horizontal/vertical margin, surgical resection of stomach should be considered. In case..."
Line 504. Substitute "supplemented" with "combined"
Line 505. Substitute the acronym MTD with "multidisciplinary team"
Line 510. Substitute "The idea is to use" with "This combination therapy consists of a ..."
Line 510. Delete "regimen"
Lines 512-513. Delete "as standard care for patients who are able to tolerate a triple cytotoxic drug regimen"
Line 514. Delete "cisplatin"
Line 519. Substitute the sentence with "... recommended; in these patients adjuvant radiotherapy may be used if they did not received D2 lymphadenectomy (94. 95). No specific..."
Lines 524-525 Substitute "and after its completion, if there is an indication for resection, surgery shoul be considered" with "followed by gastrectomy if the tumour would become operable"
Lines 531-540. Substitute with: "Peritoneal diffusion is common in GC and produces a very short life expectancy (3-6 months). In limited peritoneal carcinomatosis, hyperthermic intraperitoneal chemotherapy (HIPEC) has been used. In a skilled team the treatment is safe, but its efficacy is limited and should be confirmed (99-102). Depending on localization of carcinomatosis, peritonectomy may be a therapeutic option able to confer a survival benefit in selected patients, alone or in combination with early post-operative intraperitoneal chemotherapy (EPIC) (103, 104). In patients with unresectable..."
Lines 590-592. This sentence should be attenuated
Lines 598-600. AA. should note that in the manuscript the sensibility and specificity of diagnostic methods have not been addressed and that efficacy of treatment in term of survival and/or symptom control have not been discussed. So, these generic and trivial conclusions should be give way to considerations more consistent with the extensive material presented.
Ref. 106. I am not sure that ref. 106 is consistent with the sentence in line 549.
Comments on the Quality of English LanguageSee above
Author Response
Dear Madam/Sir,
Thank you for the favourable review and comments We did our best to improve our manuscript. Below we enclose brief answers to yours comments:
I appreciate AA. efforts to comply with previous suggestions, but a number of criticisms remains and, in my opinion, some roughness and convolution worsens understanding and fluency of the manuscript.
Here, I report some further suggestions.
- Heading of figure 2. Substitute "onsets" with "onset"
We did it.
- Line 313. Add "Immune checkpoint inhibitors in combination with standard therapy improve OS in both HER2 negative and HER2 positive GC according with KEYNOTE-811, CheckMate-649 and KEYNOTE-859 trials" (ref. Janjigian YY Lancet 2023 and JCO 2024; Sun Young Kha Lancet)
We did it.
- Line 435. I am not sure that Ca 15.3 may be used as a biomarker in GC
We did it.
- Lines 415-418. I suggest to reverse the statements: "Carbohydrate antigen Ca 19.9 (Ca 19.9) is the most commonly used serum tumour marker for GC. However, it may be overexpressed also in pancreatic cancer as well as in other gastro-intestinal tumours (74). In GC Ca 19.9 is mainly useful in checking for relapse and monitoring metastatic disease. The usefulness ..."
We did it.
- Line 439. Delete "as a strategy"
We did it.
- Lines 439-445. Substitute with: "Gastrectomy with D2 lymph node dissection is the main surgical procedure with intention of cure. In tumours localized in the mid-distal part of stomach, distal gastrectomy preserving the upper third of stomach is considered equivalent to total gastrectomy, which is reserved to proximal tumours. Preferable proximal margin of resection ..."
We did it.
- Lines 448-449. Delete or explain better the sentence "In the case of tumours located in the proximal part of the stomach, upper resection is possible"
We deleted the lines 448-449.
- Line 450. Substitute "it is preferable to ensure R0 resection" with "R0 resection should be ensured"
We did it.
- Line 455. Pospone this sentence at the end of the paragraph
We did it.
- Lines 456-458. Substitute with "Routine splenectomy is not recommended but it should be performed if spleen or its hilar lymph nodes are invaded"
We did it.
Line 463. Substitute with "Palliative, non-radical, interventions may be used in particular circumstances. Gastric..."
We did it.
- Line 464. Substitute "a cancerous tumour" with "GC"
We did it.
- Line 467. Delete "cancer"
We did it.
- Lines 469-470. Substitute "...it is allowed to resect the primary tumour along with metastases if it is possible to achieve R0" with "R0 resection of the primary tumour along with metastases is allowed"
We did it.
- Lines 475-476. Delete "which is emphasized by many randomized controlled trials in progress"
We did it.
- Line 476. Substitute "peripheral surgery such as laparoscopic resection" with "laparoscopic distal gastrectomy is considered equivalent to laparatomy..."
We did it.
- Lines 494-498 The clinical sense of the sentences is unclear without a larger explanations. In the context of the present review I think better substitute them with "Completeness of the primary tumour removal after EMR/ESD must be evaluated according to JGCA criteria for endoscopic curability, that are different for tumours with dominant diffuse or undifferentiated type (ref. https://doi.org/10.1007/s10120-020-01042-y). Whether radicality of resection is not sure, in particular when the lesion is not resected en-bloc or had a positive horizontal/vertical margin, surgical resection of stomach should be considered. In case..."
We did it.
Line 504. Substitute "supplemented" with "combined"
We did it.
Line 505. Substitute the acronym MTD with "multidisciplinary team"
We did it.
Line 510. Substitute "The idea is to use" with "This combination therapy consists of a ..."
We did it.
Line 510. Delete "regimen"
We did it.
Lines 512-513. Delete "as standard care for patients who are able to tolerate a triple cytotoxic drug regimen"
We did it.
Line 514. Delete "cisplatin"
We did it.
Line 519. Substitute the sentence with "... recommended; in these patients adjuvant radiotherapy may be used if they did not received D2 lymphadenectomy (94. 95). No specific..."
We did it.
Lines 524-525 Substitute "and after its completion, if there is an indication for resection, surgery shoul be considered" with "followed by gastrectomy if the tumour would become operable"
We did it.
Lines 531-540. Substitute with: "Peritoneal diffusion is common in GC and produces a very short life expectancy (3-6 months). In limited peritoneal carcinomatosis, hyperthermic intraperitoneal chemotherapy (HIPEC) has been used. In a skilled team the treatment is safe, but its efficacy is limited and should be confirmed (99-102). Depending on localization of carcinomatosis, peritonectomy may be a therapeutic option able to confer a survival benefit in selected patients, alone or in combination with early post-operative intraperitoneal chemotherapy (EPIC) (103, 104). In patients with unresectable..."
We did it.
Lines 590-592. This sentence should be attenuated
We did it.
Lines 598-600. AA. should note that in the manuscript the sensibility and specificity of diagnostic methods have not been addressed and that efficacy of treatment in term of survival and/or symptom control have not been discussed. So, these generic and trivial conclusions should be give way to considerations more consistent with the extensive material presented.
We deleted the lines 598-600.
Ref. 106. I am not sure that ref. 106 is consistent with the sentence in line 549.
We checked it and substituted ref. 106 with 109.
Round 3
Reviewer 1 Report
Comments and Suggestions for Authors
AA carried out all the changes that I suggested and have carefully revised the English language.